Automated analysis of small RNA datasets with RAPID

Karunanithi Sivarajan 1 2 3
Simon Martin 4
Schulz Marcel H. marcel.schulz@em.uni-frankfurt.de 1 3 5
1 Cluster of Excellence for Multimodal Computing and Interaction, and Department for Computational Biology & Applied Algorithms, Max Planck Institute for Informatics, Saarland Informatics Campus , Saarbrücken , Germany
2 Graduate School of Computer Science, Saarland Informatics Campus, Universität des Saarlandes , Saarbrücken , Germany
3 Institute for Cardiovascular Regeneration, Goethe University Hospital , Frankfurt am Main , Germany
4 Molecular Cell Biology and Microbiology, Wuppertal University , Wuppertal , Germany
5 German Centre for Cardiovascular Research (DZHK), Partner site RheinMain , Frankfurt am Main , Germany
Peña-Castillo Lourdes
Electronic publication date: 2019 Apr 10
Publication date: 2019
Volume: 7
Electronic Location ID: e6710
Received 2018 Nov 29; Accepted 2019 Mar 1
Copyright: ©2019 Karunanithi et al.
Copyright year: 2019
Copyright holder: Karunanithi et al.
License: This is an open access article distributed under the terms of the Creative Commons Attribution License, which permits unrestricted use, distribution, reproduction and adaptation in any medium and for any purpose provided that it is properly attributed. For attribution, the original author(s), title, publication source (PeerJ) and either DOI or URL of the article must be cited.
License URL: https://creativecommons.org/licenses/by/4.0/

Keywords: Comparative analysis, sRNA tool, Automated sRNA analysis, sRNA, Small RNA analysis, Computational sRNA analysis, siRNA analysis, siRNA quantification, Eukaryotic sRNA

Funding: German Research Council (DFG) SI1379/3-1 SCHU3140/1-1 The Max Planck Society This work was supported by grants from the German Research Council (DFG) to Martin Simon (SI1379/3-1) and to Marcel H. Schulz (SCHU3140/1-1). The Max Planck Society funded the Article Processing Charge (APC). The funders had no role in study design, data collection and analysis, decision to publish, or preparation of the manuscript.

==============================
Understanding the role of short-interfering RNA (siRNA) in diverse biological processes is of current interest and often approached through small RNA sequencing. However, analysis of these datasets is difficult due to the complexity of biological RNA processing pathways, which differ between species. Several properties like strand specificity, length distribution, and distribution of soft-clipped bases are few parameters known to guide researchers in understanding the role of siRNAs. We present RAPID, a generic eukaryotic siRNA analysis pipeline, which captures information inherent in the datasets and automatically produces numerous visualizations as user-friendly HTML reports, covering multiple categories required for siRNA analysis. RAPID also facilitates an automated comparison of multiple datasets, with one of the normalization techniques dedicated for siRNA knockdown analysis, and integrates differential expression analysis using DESeq2.

Availability and Implementation

RAPID is available under MIT license at https://github.com/SchulzLab/RAPID. We recommend using it as a conda environment available from https://anaconda.org/bioconda/rapid

Introduction

Widespread availability of small RNA (sRNA) sequencing technologies drives the biological community in unraveling the pivotal role of sRNA molecules. Micro RNA (miRNA), short interfering RNA (siRNA), piwi-interacting RNA (piRNA), small nucleolar RNA (snoRNA), and trans-acting RNA (taRNA) are some members of the sRNA family. In a wide range of organisms, these sRNA molecules play crucial roles in gene regulation (Bossi & Figueroa-Bossi, 2016). Although miRNAs are the most widely studied sRNA molecules, a growing interest can be seen in other sRNA classes, like siRNAs. With improved mechanistic understanding of siRNA function, siRNAs are increasingly used as therapeutic agents in drug discovery (Chavan-Gautam, Shah & Joshi, 2017). Using siRNAs in therapy requires a solid understanding of siRNA biogenesis, and behavior.

Understanding siRNA biogenesis and function often involves the computation of various sequence properties like length, strand of origin, and soft-clipped nucleotides of sRNA molecules. A myriad of available sRNA analysis tools substantiate the complexity in analyzing sRNA data sets. Existing sRNA analysis tools can be broadly categorized into two categories based on their function. (i) Tools which are dedicated to predict novel miRNAs, piRNAs, etc. using diverse computational strategies. This list includes methods such as Shortstack (Axtell, 2013), miRDeep2 (Friedländer et al., 2012), iMir (Giurato et al., 2013), Piano (Wang et al., 2014), etc. (ii) The secondary focus of many sRNA analysis tools is to annotate, and perform Gene Ontology (GO) enrichment analysis of known, or predicted sRNAs. Examples include miRTools2 (Wu et al., 2013), iSmart (Panero et al., 2017), and CPSS (Wan et al., 2017). However, such annotation based tools lack user-flexibility as they are hardcoded to work only in certain genomes like humans or mouse primarily. This hampers researchers working with uncommon model organisms. Only very few tools, like sRNAtoolbox (Rueda et al., 2015), Oasis (Capece et al., 2015), and ncPRO-Seq (Chen et al., 2012), do not have a hard-coded genome constraint, but they lack diverse graphical representation of data. In addition, existing tools are not tailored to compare multiple samples in a systematic way, properly normalizing sRNA datasets, thus allowing for an unbiased analysis. A non-exhaustive list of available sRNA analysis tools, and their abilities in addressing various properties essential to understand sRNA biogenesis, and mechanisms are discussed in Table 1. In spite of the diverse availability of sRNA analysis tools, they have potential mishaps and do not capture all the qualitative, and quantitative properties (discussed in Tables 1 and 2) while equipping the user with unbiased multi-sample comparisons.

Table 1 Comparison of RAPID with other tools is shown.

√, Feature supported; x, Feature not supported; NA, Feature is not in the scope of this tool. For instance, Knockdown corrected normalization feature is NA for CPSS, because it does not support multiple sample comparison. The full description of the column headers are listed in Table 2.

							Qualitative plots	Quantitative plots						
Tool/supporting feature	Contaminant removal	Supports other aligners	User-defined gene/region	Knockdown corrected normalization	Offline	Hardcoded genomes	Heatmaps	PCA	MDS	Strand specificity	Soft-clipped bases	Coverage plots	Multi-sample comparison plots	Differential analysis	Enrichment analysis support	Interactive interface	miRNA or piRNA specific?	
RAPID	√	√	√	√	√	x	√	√	√	√	√	√	√	√	x	x	x	
smallRNA toolkit (Moxon et al., 2008)	x	x	√	NA	x	√	x	x	x	x	x	x	x	x	x	x	√	
sRNA toolbox (Rueda et al., 2015)	x	x	√	NA	√	x	x	x	x	x	x	x	x	√	√	x	√	
Oasis (Capece et al., 2015)	x	x	x	x	x	x	√	√	x	x	x	x	√	√	√	√	√	
CPSS (Wan et al., 2017)	x	√	x	NA	x	√	x	x	x	x	x	x	x	x	√	√	√	
iSmart (Panero et al., 2017)	x	x	x	NA	√	√	√	√	x	x	x	x	x	√	√	√	√	
iSRAP (Quek et al., 2015)	x	√	x	NA	√	x	√	√	√	x	x	x	x	√	x	x	x	
PiPipes (Han et al., 2015)	√	√	x	NA	√	x	x	x	x	√	√	x	x	√	x	x	√	
ncPRO-Seq (Chen et al., 2012)	x	√	√	NA	√	x	x	x	x	x	x	x	x	x	√	x	√	
UEA sRNAworkbench (Mohorianu et al., 2017)	√	√	√	x	√	x	x	x	x	x	x	x	√	√	x	√	x	
NGSToolbox (Rosenkranz et al., 2015)	x	x	√	NA	√	x	x	x	x	x	x	x	x	x	x	x	√	
SePIA (Icay et al., 2016)	x	√	x	NA	√	x	x	x	x	x	x	x	x	√	x	x	√	
SPAR (Kuksa et al., 2018)	x	√	x	NA	x	√	√	x	x	x	x	x	x	x	x	√	√	
Notes.

Tools whose primary focus is on identifying/annotating different classes of small RNAs are not included.

Table 2 Table describing the supporting features of RAPID (Column headers in Table 1).

Supporting feature	Description	
Contaminant removal	Is there an option to remove set of contaminants (microbial, ribosomal, etc.) from the read files?	
Supports other aligners	Does the tool support alignment files from other tools, instead of performing their own alignment?	
User-defined gene/region	Could the user specify a list of regions to perform downstream analysis?	
Knockdown corrected normalization	Does the tool enable multiple-sample comparison by facilitating normalization techniques specific to sRNA knockdown studies?	
Offline	Can the tool be used offline?	
Hardcoded genomes	Is the tool generic? i.e., Is the tool’s ability somehow limited to a set of pre-defined genomes?	
Quantitative, and Qualitative Plots	Does the tool support informative plots to gain understanding of the analyzed data (MDS=multi-dimensional scaling, PCA= principal component analysis)	
Multi-sample comparison plots	Does the tool provide a comprehensive view of multiple samples (not just differential analysis)? For instance, how does the read distribution vary across multiple samples in different genes of interest?	
Differential analysis	Is the tool equipped with modules to perform pairwise differential analysis?	
Enrichment analysis support	Is there any support to perform functional enrichment within the tool	
Interactive interface	Does the tool have an interactive interface, or plots?	
miRNA or piRNA specific?	Is the tool specific to analyze miRNA or piRNA only?	

Hence, we developed a generic sRNA analysis offline tool: Read Alignment, Analysis, and Differential PIpeline (RAPID), primarily tailored to investigate eukaryotic siRNAs. RAPID quantifies the basic alignment statistics with respect to read length, strand bias, non-templated nucleotides, nucleotide content, sequencing coverage etc. for user-defined sets of genes or regions of any reference genome. Once basic statistics are computed for multiple sRNA datasets, our tool aids the user with versatile functionalities, ranging from general quantitative analysis to visual comparison of multiple sRNA datasets.

Materials and Methods

Figure 1 shows an overview of the various modules of RAPID, which we discuss below.

Figure 1 The Pipeline of our tool RAPID is depicted. Green boxes are executables; blue, and orange boxes represent input, and output files respectively.

The executable RAPID modules are: (i) rapidStats module performs reference alignment and quantifies the expression of user-defined genes and/or regions. (ii) rapidNorm facilitates sample (or gene) wise comparison of genes/regions (or samples) after appropriate normalization. (iii) The rapidVis module provides multiple visualizations representing the information obtained from rapidStats and rapidNorm. Selective screenshots from the output of our case studies are shown in the boxes. (iv) rapidDiff is the differential expression analysis module implementing DESeq2.

Basic module

The first of four RAPID modules is rapidStats, which performs sequence (FastQ) alignment, with or without contaminant removal, using Bowtie2 (Langmead, Salzberg & Langmead, 2013). After alignment, RAPID obtains read statistics such as read length distribution, soft-clipped nucleotides, strandedness, and nucleotide content. RAPID can skip the alignment and directly use alignment files (BAM/SAM) as well. To efficiently process, capture and store the aforementioned statistics, RAPID uses SAMtools (Li et al., 2009), BEDtools (Quinlan & Hall, 2010), and custom Perl, Shell, and R scripts. The statistics captured by this module serve as input for other modules.

Normalization module for multi-sample comparison

RAPID aims to facilitate an unbiased comparison of genes or regions across multiple sRNA samples. Other than the sequencing depth itself, sRNA studies pose an additional challenge during normalization. For instance, to understand RNA interference (RNAi) mechanisms and how the siRNA homeostasis is maintained, often a gene or siRNA region is knocked down. One such knockdown strategy is to introduce large amounts of siRNAs, called primary siRNAs, against the knockdown gene or any siRNA region. Consequentially, secondary siRNA production is triggered by the primary siRNAs. These primary and secondary siRNAs, which are also sequenced, can add up to millions of reads in the total library size.

To our knowledge, there are no normalization methods specialized for knockdown based small RNA-seq studies. However, many methods have been proposed to normalize mRNA-seq data, which can be broadly categorized in two classes: (i) total count scaling (TCS) methods and (ii) methods which utilize quantities like median log-fold change, among all genes between mRNA-seq experiments. To be able to use the latter methods, sRNA loci annotation should be available, and should assume that most of the sRNA loci between samples are not differentially expressed. In model organisms like Paramecium tetraurelia, little is known about the localization, and expression variability of endogenous small RNA loci. Hence, the second class of methods may not be applicable. However, the disadvantage of TCS methods is that the used normalization factors were shown to be biased by highly expressed genes in the dataset (Dillies et al., 2013). In case of knockdown samples, TCS methods will be heavily biased because of the millions of primary, and secondary siRNAs associated with the knockdown gene or region.

In mRNA-seq data, a variant of TCS method (Sultan et al., 2008) was introduced, where normalization is achieved by scaling through a factor that estimates the difference in the number of reads mapped between samples. We previously proposed a variant of the TCS method in a knockdown based siRNA study (Götz et al., 2016). Here, we term this variant as KnockDown Corrected Scaling (KDCS) method, where we remove from the estimated total library size, all small RNA reads that map against the knockdown genes, this quantity is denoted as K below. Assume read count R for a region of interest that we want to compare between samples. T is the total number of reads mapping to the genome, and K is the number of small RNA reads mapping to the knockdown gene. We compute the normalized read count R ˆ: (1) R ˆ=R⋅MT−K,

where M is the maximum over all values (T1 − K1), …, (Tn − Kn) over all n samples. RAPID uses the KDCS method, by default. Hence, in the absence of knockdown genes, the normalization works as the normal TCS method. However, in order to provide flexibility with the choice of normalization for knockdown free analysis,we have also incorporated size factor-based normalization from DESeq2 (Love, Huber & Anders, 2014). If an user can safely assume that most of the genes or regions between samples are not differentially expressed, in a small RNA based study, then they can use the DESeq2 normalization.

Visualization module

As visualization enables better understanding of data, the rapidVis module of RAPID automatically generates insightful plots from the output of previous modules. RAPID makes use of Rmarkdown (http://rmarkdown.rstudio.com) to create easily navigable HTML reports. This module contains two modes: statistics and comparison mode. The statistics mode accepts input from the rapidStats output file, and provides various single category plots detailing on the distribution of read length, strandedness, soft-clipped nucleotides, and coverage plots for each gene/region analyzed. In addition, this report also provides combinations of the aforementioned properties. For instance, how does strandedness differ across different read lengths. Comparison mode accepts the rapidNorm analysis output file, to equip the user with qualitative reports (Heatmaps, PCA, MDS) of samples. Further, sample and gene/region wise comparison plots of the properties inherent in the data. All plots are shown both in normal and log scale such that the user can directly incorporate them into publications.

Differential analysis module

Differential Expression (DE) analysis is one of the common downstream analysis in comparative studies. RAPID equips the user with this functionality by incorporating the DESeq2 package. Upon invoking the rapidDiff module, raw counts are utilized from the output of the rapidStats module to perform DE analysis, with default parameters of DESeq2. Results of the DE analysis include intuitive plots (such as MA Plot, Heatmap, PCA) and the list of DE genes/regions.

Usage and availability

We strongly recommend using RAPID from https://anaconda.org/bioconda/rapid as a conda recipe. However, it can also be freely accessed from https://github.com/SchulzLab/RAPID. A detailed use case based documentation is provided at http://rapid-doc.readthedocs.io/en/latest/.

Definitions

Here we describe the formula used in the case studies.

Coefficient of variation (CoV): Coefficient of variation is the ratio of standard deviation to mean of the data set. For a gene or region of interest, i, with n samples (2) CoVi=σiμi,

where σi, and μi are the standard deviation, and mean of the gene or region of interest i in the n samples respectively.

AntiSense Ratio (ASR):

Antisense ratio is the ratio of the number of antisense reads to the total number of reads in a gene or region. If R, and AS are the total, and antisense read counts of a region of interest, i, respectively, antisense ratio is calculated as follows: (3) ASRi=ASiRi.

DatasSets

We show the application of RAPID, using two different datasets which are briefly described below.

Paramecium tetraurelia

We used four small RNA sequencing data sets (European Nucleotide Archive (ENA) accession: PRJEB25903) from the wildtype serotypes (51A, 51B, 51D, and 51H) of P.tetraurelia. We performed adapter-trimming, merged the replicates, and extracted reads of length 21-25nt only from each dataset for this analysis. We analyzed only the rDNA cluster producing 17S, 5.8S, 25S ribosomal RNAs, External Transcribed Spacer (ETS), Internal Transcribed Spacer 1(ITS1), and Internal Transcribed Spacer 2 (ITS2). The rDNA cluster sequence can be obtained from GenBank accession: AF149979.1 (Preer et al., 1999), with the additional annotation of the 5.8S sequence from GenBank accession: AM072801.1 (Barth et al., 2006).

In addition, to demonstrate a simple effect of KDCS normalization, we utilized the five available ICL knockdown data sets from this study (NCBI accession ID: PRJEB13116) (Götz et al., 2016). After preprocessing the data as mentioned in the study, we chose four small RNA regions as examples (these are regions of the ND169 gene, as shown in the study) to quantify, and compare their sRNA accumulation across samples using RAPID.

Schizosaccharomyces pombe

We explored the 24 h time point datasets of WT, and three different knockdowns of S.pombe. The respective data sets can be obtained by the accession IDs: GEO: GSE89151; GSM2359756—wt_24 h; GSM2359762—ago1D_24 h; GSM2359768—clr4D_24 h; GSM2359774—dcr1D_24 h. We processed the data as mentioned in the corresponding download pages, before subjecting them to RAPID. We restricted our analysis to the sRNA-enriched genes available from their supplement (Joh et al., 2016) which can be accessed from https://bit.ly/2GSLcks. After pre-processing we subjected each GSM dataset to rapidStats, and compared them using rapidNorm.

Results

We show the application of RAPID on two different datasets, highlighting some features that can be investigated by doing a standard RAPID analysis.

Comparison of Paramecium tetraurelia serotypes

P. tetraurelia is a unicellular, free-living ciliate commonly found in fresh-water lakes. They show nuclear dimorphism and have a wide range of phenotypes. One such phenotype that depends on an epigenetically controlled, mutually exclusive expression of members of a multigene family is called serotypes (Cheaib et al., 2015). With a dimorphic nucleus and more than 11 serotypes, P. tetraurelia sRNAs are involved in regulatory mechanisms at the post-transcriptional, and epigenetic level (Götz et al., 2016; Cheaib et al., 2015; Carradec et al., 2015).

Our first example is an analysis on four sRNA-seq datasets (ENA: PRJEB25903) from wildtype serotypes (51A, 51B, 51D, and 51H) of P. tetraurelia. We were interested in sRNAs produced in the rDNA cluster producing 17S, 5.8S, 25S ribosomal RNAs. A simple genomic visualization of the different components of rDNA cluster regions we quantified can be seen in Fig. 2. We can observe from the strand-specific read distribution plots in Fig. 2 that the regions, namely External Transcribed Spacer (ETS), Internal Transcribed Spacer 1(ITS1), and Internal Transcribed Spacer 2 (ITS2), which get excised in the processing of polycistronic pre-rRNA, accumulate 23nt antisense small RNAs. It is known from yeast, that rRNA maturation involves co-transcriptional endonucleolytic cleavage and highly concerted trimming events to subsequently process the final rRNAs (Henras et al., 2015). Our data here suggests that in P. tetraurelia these elimination processes are associated with antisense siRNAs, possibly produced from RNA-dependent RNA Polymerase activity.

Figure 2 Small RNAs map to different units of the ribosomal DNA of P.tetraurelia.

(A) Genomic localization of different ribosomal units (Block arrows, with respective start positions) is shown. (B) The read length distribution of small RNAs mapping to different regions of the ribosomal DNA is shown. Each row corresponds to each wildtype serotype (51A, 51B, 51D, 51H respectively). Each column corresponds to ribosomal DNA regions in order: External Transcribed Spacer (ETS), 17S, Internal Transcribed Spacer 1(ITS1), 5.8S, Internal Transcribed Spacer 2 (ITS2), and 25S.

Analysis of our case studies, and their respective figures can be reproduced with the help of ( Data S1).

Normalization case study in Paramecium tetraurelia knockdowns

One of the unique features of RAPID is the KDCS normalization that can correct for the excess of sRNAs introduced in knockdown experiments in experimental approaches utilized in many diverse organisms. To demonstrate the effect of KDCS normalization, we utilized the ICL knockdown data sets from the study by Götz et al. (2016). This study investigates the molecular mechanisms of different sets of trans-acting RNAi components in P. tetraurelia. ICL is a gene in P. tetraurelia, which is not involved in the RNAi machinery. In the original study, as a control, the ICL gene is knocked down by introducing primary siRNAs against ICL (see ‘Methods’). In our work, we quantified the sRNA read counts of four example sRNA regions (which are in the original study different constructs of the ND169 gene) from the mentioned datasets. In this setup we expect that all datasets behave the same, as these are biological replicates of the same system.

As described earlier, very little is known about the localization, and expression variability of endogenous small RNA loci in P. tetraurelia. Therefore, in these sRNA knockdown samples, normalization methods, such as DESeq2, may be inappropriate, due to the assumption that the majority of regions are unchanged. We compared the effect of the TCS, DESeq2 and KDCS normalization approaches to using no normalization. We used the coefficient of variation (CoV) to measure the performance of the normalization method (see Methods; Eq. (2)) as normalization should reduce the variance in read count per region. A smaller CoV suggests a better performance of the normalization method.

Figure 3 shows the CoV values of the raw, and normalized sRNA read counts, for four example regions that had been studied by Götz et al. (2016). We can observe from Fig. 3, that the KDCS method performs better in all the regions, compared to the generic TCS method. It also performs as good or slightly better than the normalization of DESeq2 for this example. All normalization approaches are better than using no normalization, which strongly argues for their use. This experiment suggests that our KDCS method is a better alternative to the TCS method and is applicable when few regions are known.

Figure 3 The effect of different normalization methods on sRNA regions (x-axis) studied in (Götz et al., 2016) are assessed using the coefficient of variation (y-axis; lower is better) of the read counts obtained from RAPID.

Raw, No normalization; KDCS, KnockDown Corrected Scaling; TCS, Total Count Scaling; DESeq2, size factor-based normalization from DESeq2.

Analysis of Schizosaccharomyces pombe knockdowns

The fission yeast, or Schizosaccharomyces pombe, is another widely studied unicellular eukaryotic model organism, where RNAi pathways are prevalent. We explored the time point datasets of WT, and three different knockdowns of S.pombe from the study by (Joh et al., 2016). In this study, the authors investigated quiescence associated changes in small RNA transcriptomes and epigenetic modifications to identify the key players involved in quiescence. They also examined the role of RNAi proteins, by knocking out three of them, namely, Ago1, Clr4, and Dcr1. One of the key findings was that during quiescence in S.pombe, a set of sRNA-enriched genes were identified as crucial elements for the survival of the organism (Joh et al., 2016). We explored these sRNA-enriched genes using RAPID, to demonstrate the discovering potential of RAPID.

With a simple RAPID analysis comparing the different knockout samples, it was easy to screen for interesting properties in the data. We discovered that a subset of these sRNA-enriched genes have relatively higher antisense ratio (ASR, see ‘Methods’ Eq. (3)) (Fig. 4). This increased ASR was observed in different sets of genes in various knockouts. The subgroup of genes with higher ASR, could play a cis-regulatory role to silence the genes. While further investigation is necessary, our RAPID analysis (Fig. 4) suggests an involvement of different sRNA mechanisms in ensuring the survival of S.pombe in quiescence.

Figure 4 A heatmap of the antisense read ratio in the sRNA enriched genes (y-axis) across all samples (x-axis) analyzed.

Accumulation of antisense sRNA can be observed in the lower part of the heatmap, and an increase in antisense sRNA can also be seen in different knockouts compared to wildtype.

Discussion

The long list of available sRNA analysis tools attributes to the complexity, and importance of sRNAs in biological studies. However, most available tools only focus on identifying, and annotating the different classes of sRNA. They fail to characterize and visually represent the multitude of parameters crucial for understanding the sRNA world.

RAPID is designed to capture the diverse eukaryotic siRNA characteristics innately found in sRNA sequencing data sets. Some of the properties captured during the basic analysis of RAPID include read length, strand bias, non-templated nucleotides, nucleotide content, sequencing coverage etc., for user-defined sets of genes (or regions) of any reference genome. With a separate module for normalization, RAPID simplifies multi-sample comparison. We have also included an alternative normalization technique, KDCS, specially designed to aid the comparison of sRNA-based knockdown studies. KDCS normalization method can also be helpful in correcting for the transcribed small RNAs from the non-insert RNA locations of a vector. For instance, in RNAi vector constructs, like L4440 in Caenorhabditis elegans, due to lack of specificity of the termination enzyme, the non-insert RNA locations will get transcribed (Saskói et al., 2018). These regions which contribute unwanted variation can be excluded by specifying them as background in the RAPID analysis.

RAPID currently addresses many features which are crucial towards the understanding of siRNA biogenesis, and function. In spite of RAPID’s diverse functionality, there are a few shortcomings. RAPID depends on user supplied set of contaminants instead of auto-detecting it from the sequence file. The visualizations provided by RAPID in the statistics mode do not include sequence level properties, like over represented sequences or sequence logos, etc. These are interesting additions for future releases. In the comparison mode of our visualization module, the plots provided are non-exhaustive. For instance, one might like to compare the nucleotide content, or strand-based distribution of genes (or regions) across multiple samples, or vice-versa. Such special features can be requested by users in GitHub, which could be incorporated in further releases.

Conclusion

RAPID is an offline, open-source, user-friendly, and automated pipeline designed to simplify data analysis, tailored to investigate eukaryotic siRNAs. RAPID is not an exhaustive sRNA analysis or annotation pipeline. With an available set of sRNA localizations, our tool can be used to analyze single or multiple sRNA samples at ease with the aid of different normalization techniques. The diverse set of visualizations generated by RAPID will enhance the understanding of any sRNA-based study. RAPID is available for free use and can be used over the command line. It is available at the github repository (https://github.com/SchulzLab/RAPID). A detailed user-tutorial can be accessed from this repository.

Supplemental Information

Supplemental Information 1 Data associated with the case studies

The script (ReproducePlots.sh) can be used to reproduce all the figures, and analysis performed in the manuscript.

Click here for additional data file.

We would like to thank Miriam Cheaib, Angela Rodrigues, and Fatemeh Behjati Ardakani for their valuable feedback.

Additional Information and Declarations

Competing Interests

Author Contributions

Data Availability

The authors declare there are no competing interests.

Sivarajan Karunanithi conceived and designed the experiments, performed the experiments, analyzed the data, contributed reagents/materials/analysis tools, prepared figures and/or tables, authored or reviewed drafts of the paper, approved the final draft.

Martin Simon conceived and designed the experiments, authored or reviewed drafts of the paper, approved the final draft, he contributed to Discussions, and data set identification for our case studies.

Marcel H. Schulz conceived and designed the experiments, contributed reagents/materials/analysis tools, authored or reviewed drafts of the paper, approved the final draft.

The following information was supplied regarding data availability:

Our tool is openly accessible at https://github.com/SchulzLab/RAPID.

Case studies explained in this manuscript can be reproduced with Data S1.

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
