# Peer review of "Automated analysis of small RNA datasets with RAPID"

_PeerJ, doi:10.7717/peerj.6710_

## Round 0.1 · original submission · Major Revisions

The reviewers found that the RAPID software is a positive contribution to the field; however, since this manuscript is about a Bioinformatics tool is imperative that the tool is easily installed by users. Thus, please fix the installation problems the reviewers encountered. Additionally, the software should be accompanied by a minimal script that downloads the data, loads it, and performs the analysis to reproduce the results (tables, plots, visualizations etc.) in the manuscript (see https://peerj.com/about/policies-and-procedures/#discipline-standards).

Please address the reviewers' comments. Specially, fix Table 1 and Figures 2 and 3 (the first is upside down and a reviewer questioned its fairness to the other tools, the second has tiny labels, and the third is missing DeSeq2 results); clarify the class of sRNA supported by RAPID, and include a complete example analysis (this could be part of the supplementary material).

·

Basic reporting

no comment

Experimental design

no comment

Validity of the findings

no comment'

Additional comments

The tool fills a gap but its current stage of the deployment is in my opinion not optimal. While the author offer an installation via the package manager conda, the package seems to be not properly set up. I personally have not created a conda package for a Shell or R script so far but I would imagine that it also for those possible to generate proper callable versions. In the current implementation the user has to set the path to the rapid file (in the documentation suggested as "rapid=/home/<username>/miniconda2/envs/<environment_name>/bin/ (or) rapid=/Users/<username>/miniconda2/envs/<environment_name>/bin/") and has to provide the command-line parameter "-r" with that path. In my opinion this is not really user-friendly and should be improved.

Further issues suggestions:

- I would recommend to submit the code also to Zenodo to ensure it availability (https://guides.github.com/activities/citable-code/) and to add a CFF file (https://citation-file-format.github.io/) for better citability.

- Please write in the manuscript that the tool is for eukaryotic sRNA. Bacterial sRNA are as far as I understand not supported by the tool but the distinction is not explicitly done.

- The Docker image does not work.
'''
$ docker pull quay.io/biocontainers/rapid
Using default tag: latest
Error response from daemon: manifest for quay.io/biocontainers/rapid:latest not found
'''

- The generated heatmaps use a red-green colormap that are not
red-green-blindness safe.

- The documentation page has no proper title tag and the file
"Installation.rst" is missing a newline. Due to this the code block
"conda search rapid" is not properly formated.

Reviewer 2 ·

Basic reporting

The paper by Karunanithi et. al describes a pipeline to visualize specific aspects of RNA sequencing data for pre-defined genomic loci. I am not an expert in the field of eukaryotic sRNAs and can only provide an outsiders view on the topic.
The manuscript is clearly written and appropriate literature is cited. The existing figures are justified, however the majority of the RAPID visualization features is not covered by a figure and not explained based on examples. It would be also informative for the reader to give more examples and guidance of how a specific RAPID output facilitate the analysis of sRNAs. For example, what is the benefit of the read length distributions? Is it possible to draw conclusions for sRNA synthesis or mode of action? As a general question; how meaningful is the length distribution for reads which are pre-filtered for a given length?
Due to the completely different scopes of the tools I think the comparison in Table 1 is kind of misleading. Nearly all of the displayed tools are miRNA or piRNA specific with many additional dedicated features not mentioned in the table, while besides in principle generic, RAPID seems to be primarily tailored for siRNAs.

Experimental design

As mentioned above the method seems to be tailored regarding to the needs of an siRNA analysis and maybe other cis-acting RNAs. This should be clearly stated within abstract and introduction.
The presented data reflect only a subset of RAPID functions. If the missing functions are not relevant they should be omitted, otherwise they should be explained in the paper. I would highly appreciate a complete example analysis based on a given locus including all visualizations and a reference discussion/interpretation of the results.
In general I am struggling to understand the term “modified bases” in this context. In my understanding modified bases are chemically modified by e.g. methylation. How can this be detected by Ilumina sequencing of the cDNAs, or does "modified bases" refer to another aspect?

Validity of the findings

The comparison of the normalization methods in Figure 3 lacks the results for the DESeq2 method.

Additional comments

Minor points:
Random searching for the tools in Table 1 showed that "sRNAanalyazer" is misspelled (sRNAnalyzer).

Reviewer 3 ·

Basic reporting

The introduction provides sufficient background and motivation for the software. There is clearly plenty of software available for small RNA analyses, and at least some of the literature on this is referenced, and a comparative table of their features has also been provided. Though I’ve only examined these aspects cursorily: the manuscript seems to be correctly structured and formatted according to the PeerJ guidelines. The language used in the manuscript is quite acceptable too. The figures are generally clear, with the exception of Figure 2's axis labels being extremely tiny - way too small to be legible (at least in the printed version of the manuscript to be reviewed). Raw data is available.

Experimental design

Software installation, testing and reproducibility:

I think experimental design/rigorous investigation isn't the primary concern in this manuscript. However, functioning of the software is certainly worth considering. So, I'm using this section to address this. Generally I would state that the motivation for the manuscript and software are sufficient and that the methods are described adequately. Regarding replication, see the comment towards the end of this section.

I attempted to install RAPID using conda as described in the documentation. I had error messages using this installation approach that caused the pipeline to abort (and also incidental warning messages too - though these may be platform specific).

bash runTest.sh
ERROR: unknown parameter "-p"
* * *
|_______ ________ _____ |
| | | | | I | \ |
| | | /\ | | I | \ |
| |____| / \ |____| I | | |
| |\ /____\ | I | | |
| | \ / \ | I | | |
| | \ / \ | I | / |
| | \ / \ | I |__/ |
| |
| -Read Alignment, Analysis, and Differential Pipeline- V 0.1 |
|____________________________________________________________________|
Usage:

./rapidStats.sh -o=/path_to_output_directory/ -f=reads.bam -ft=BAM --remove=no --annot=file.bed --index=/path_to_index
Parameters:
-h|--help
-o|--out=/path_to_output_directory/ : path to the output directory, directory will be created if non-existent
-f|--file=filename : the input file
-ft|--filetype = BAM/SAM/fq : Mention either BAM/SAM or FASTQ. Default FASTQ
-a|--annot=file.bed : bed file with regions that should be annotated with read alignments (Multiple Bed files should be separated by commas)
-r|--rapid=PATH/ : set location of the rapid installation bin folder (e.g. /home/software/RAPID/bin/) or put into PATH variable
-i|--index=PATH/ : set location of the bowtie2 index for alignment
--contamin=yes : use a double alignment step first aligning to a contamination file (default no)
--indexco=PATH/ set location of the contamination bowtie2 index for alignment (only with contamin=yes)
--remove=yes : remove unecessary intermediate files (default yes)
Run comparative analysis using config file test.config
During startup - Warning messages:
1: Setting LC_COLLATE failed, using "C"
2: Setting LC_TIME failed, using "C"
3: Setting LC_MESSAGES failed, using "C"
4: Setting LC_MONETARY failed, using "C"

Attaching package: ‘gplots’

The following object is masked from ‘package:stats’:

lowess

Error in file(file, "rt") : cannot open the connection
Calls: read.table -> file
In addition: Warning message:
In file(file, "rt") :
cannot open file './TestRapid/Regions/alignedReads.sub.compact': No such file or directory
Execution halted
During startup - Warning messages:
1: Setting LC_COLLATE failed, using "C"
2: Setting LC_TIME failed, using "C"
3: Setting LC_MESSAGES failed, using "C"
4: Setting LC_MONETARY failed, using "C"

Attaching package: ‘gplots’

The following object is masked from ‘package:stats’:

lowess

Error in file(file, "rt") : cannot open the connection
Calls: read.table -> file
In addition: Warning message:
In file(file, "rt") :
cannot open file './TestCompare/NormalizedValues.dat': No such file or directory
Execution halted


I was wondering if I had the latest version installed, since the banner from the test command indicates version 0.1? The command “conda search rapid” gave the following:
rapid 0.7 pl526r341_0 bioconda

Can the authors fix this issue with the info I provided? My main reason for suggesting a "minor revision" is to request that this can be sorted out, as the conda installation should be the preferred option.

Due to the problems above, I switched to installation from the GitHub master branch source for RAPID. It took a while to get all the prerequisites installed, but when I did so, I believe the pipeline ran to completion, as I was able to see graphs for all the analyses. I discovered that in addition to the dependencies noted in the documentation online, RMarkdown needs to be installed too (and was described as being used in the methods).

I was not able to install pandoc as described in the software documentation (I looked around briefly on the internet to figure out where it's available, but gave up trying to determine how to get it installed in R):

> install.packages(c("pandoc"))
Warning message:
package ‘pandoc’ is not available (for R version 3.5.1)

The PeerJ standards for bioinformatics software tools (see relevant section at: https://peerj.com/about/policies-and-procedures/#discipline-standards), recommends: “If available as a package (e.g. Python, R, Matlab, Octave), it should be accompanied by a minimal script that downloads the data (if not included in the package), loads it, and performs the analysis to reproduce the results (tables, plots, visualizations etc.) in the manuscript.” Would it be too onerous to include at least some of this? This might assist in making the software more robust, and would, I think, inspire more confidence that everything is working correctly than the provided test scripts. (Alternatively, it would be helpful to have the output of the test scripts downloadable from GitHub to compare to).

Validity of the findings

Since this is primarily a software article, I think the results demonstrate the utility of the tool as necessary. Reporting on the capabilities of the software is balanced. Two relevant biological data sets were analysed. I don’t think any of the interpretations of the analyses were unduly speculative.

Additional comments

I think this software will be of utility to researchers interested in analyzing small RNAs, and look forward to the potential expansion to include sequence level analyses in future.

The normalisation for sRNA counts will be useful in cases where primary (knockdown) and secondary siRNAs comprise a large fraction of the total sRNA population. I think it would be useful if the normalisation took into account not only siRNAs against the gene to be knocked down, but also those generated from transcription of the vector backbone (both could be included in the quantity “K”). One of the common vector backbones used in RNAi (L4440) has significant amounts of non-insert RNA production due to the lack of transcription terminators (see, e.g., Sturm et al. 2018: https://academic.oup.com/nar/article/46/17/e105/5040046). A note about this issue may be useful to users of the software.

Minor comment:

Pg. 6, line 192: What is “goodness” of the normalisation method? Maybe “performance”?

---

## Round 0.2 · accepted · Accept

In my view, the authors have addressed all of the reviewers' concerns, solved the issues with the installation of their software, and provided a script that downloads the data, loads it, and performs the analysis to reproduce the results in the manuscript. Additionally, the user manual at the readthedocs website provides a comprehensive description of RAPID's installation, usage and functionalities.

My only suggestion is to change this sentence in line 201 "It also performs as good or better than..." to "It also performs as good or slightly better than..." as based on figure 3 the performance of both methods is very similar.

#